# Physicochemical Characterization and Antitumor Activity of Fucoidan and Its Degraded Products from *Sargassum hemiphyllum* (Turner) C. Agardh

**DOI:** 10.3390/molecules28062610

**Published:** 2023-03-13

**Authors:** Baozhen Luo, Zhuo Wang, Jianping Chen, Xuehua Chen, Jiarui Li, Yinghua Li, Rui Li, Xiaofei Liu, Bingbing Song, Kit-Leong Cheong, Saiyi Zhong

**Affiliations:** 1Guangdong Provincial Key Laboratory of Aquatic Product Processing and Safety, Guangdong Provincial Engineering Technology Research Center of Seafood, Guangdong Province Engineering Laboratory for Marine Biological Products, Key Laboratory of Advanced Processing of Aquatic Product of Guangdong Higher Education Institution, Guangdong Provincial Science and Technology Innovation Center for Subtropical Fruit and Vegetable Processing, College of Food Science and Technology, Guangdong Ocean University, Zhanjiang 524088, China; 13413650541@163.com (B.L.); wangzhuo4132@outlook.com (Z.W.); 18977740415@163.com (X.C.); rain8413@163.com (J.L.); liruihn@163.com (R.L.); liuxf169@126.com (X.L.); 15891793858@163.com (B.S.); klcheong@stu.edu.cn (K.-L.C.); 2Collaborative Innovation Center of Seafood Deep Processing, Dalian Polytechnic University, Dalian 116034, China; 3Center Laboratory, Guangzhou Women and Children’s Medical Center, Guangzhou Medical University, Guangzhou 510120, China; liyinghua@gzhmu.edu.cn

**Keywords:** *Sargassum hemiphylla*, separation and purification, oxidative degradation, low molecular weight fucoidans, antitumor

## Abstract

Fucoidan has many biological functions, including anti-tumor activity. Additionally, it has been suggested that low-molecular-weight fucoidans have greater bioactivities. This study aimed to examine the degradation, purification, physicochemical characterization and in vitro antitumor activity of fucoidan from *Sargassum hemiphyllum* (Turner) C. Agardh. Fucoidan was isolated using DEAE-cellulose-52 (F1, F2), Vc-H_2_O_2_ degration, and Sepharose CL-6B gel (DF1, DF2) from crude Sargassum fucoidans. Physicochemical characteristics of four isolated fucoidans were examined using chemical and monosaccharide composition, average molecular weight (Mw), and FTIR. Furthermore, the anti-proliferative effects of purified fucoidans on human hepatocellular carcinoma cells (HepG2), human Burkitt Lymphoma cells (MCF-7), human uterine carcinoma cells (Hela) and human lung cancer cells (A549) were analyzed by MTT method. The apoptosis of HepG2 cells was detected by flow cytometry. Our data suggest that the contents of polysaccharide, L-fucose and sulfate of DF2 were the highest, which were 73.93%, 23.02% and 29.88%, respectively. DF1 has the smallest molecular weight (14,893 Da) followed by DF2 (21,292 Da). The four fractions are mainly composed of fucose, mannose and rhamnose, and the infrared spectra are similar, all of which contain polysaccharide and sulfate characteristic absorption peaks. The results of MTT assay showed that the four fractions had inhibitory effects on HepG2 and A549 in the range of 0.5–8 mg/mL, and the four fractions had strong cytotoxic effects on HepG2 cells. DF2 had the best inhibitory effect on HepG2 (IC_50_ = 2.2 mg/mL). In general, the antitumor activity of Sargassum fucoidans is related to the content of L-fucose, sulfate and molecular weight, and Sargassum fucoidan has the best inhibitory effect on HepG2 hepatocellular carcinoma cells. Furthermore, when compared to MCF-7, Hela, and A549 cells, Sargassum fucoidans had the best capacity to reduce the viability of human hepatocellular carcinoma cells (HepG2) and to induce cell apoptosis, proving itself to have a good potential in anti-liver cancer therapy.

## 1. Introduction

Seaweed is an important marine biological resource, and plays an important role in the material cycle of the marine ecosystem and the maintenance of marine ecological balance. Recently, seaweed has been considered as a promising new biomass feedstock for the production of biochemicals and the development of commercial value-added products [1]. Sargassum is a type of brown algae that contains a variety of bioactive compounds, including fatty acids, polyphenols, peptides, and polysaccharides [2]. In many parts of Asia, Sargassum, as an edible brown algae, contains unique hydrocolloid polysaccharides, such as alginate and carrageenan, which is a potential source of bioactive polysaccharides [3]. Previous studies [4] on sargassum mainly focus on the structural analysis and biological activity evaluation of its main component, fucoidan. Fucoidan is a water-soluble heteropolysaccharide mainly compose of L-fucose and sulfate groups, containing uronic acid and other monosaccharides, such as glucose, xylose, galactose and mannose [5]. Its composition depends on the source of algae, extraction and purification methods, harvest time and location [6]. In addition, the structural differences of known fucoidan also affect the activity, including not only the molecular weight (M_W_) and the degree of sulfation (DS), but also sulfate linkages (i.e., 2–O, 3–O and 4–O sulfates) [7]. Consequently, each new pure fucoidan extracted from seaweed is a new compound with unique properties and structure, and has potential biological activity.

Compared to synthetic drugs, natural products as antitumor drugs have caught increasing attention for their biological activity and limited side effects. Several studies have reported that fucoidan in Sargassum can inhibit the growth of a variety of cancer cells, but not of normal cells [8]. Fucoidan from *Sargassum polycystum* has a strong antitumor effect on MCF-7 cell line with IC_50_ at 50 μg/mL [9]. Likewise, 200 μg/mL fucoidans isolated from *Sargassum thunbergii* can inhibit the growth of human colon cancer cells (DLD-1, HT-29, and HCT-116) [10]. Ye et al. [11] reported that fucoidan extracted from *Sargassum pallidum* has anti-proliferative activity on human hepatocellular carcinoma (HepG2), human lung adenocarcinoma epithelial (A549) and human gastric carcinoma (MGC-803) cells. The cytotoxic effect of fucoidan from Sargassum horneri on DLD-1 colon cancer cell line were found at concentrations from 0.5 to 6.0 mg/mL [12]. In addition, Suresh [13] evaluated the anti-tumor activity of fucoidan from different parts of *Sargassum plagiophyllum* C. Agardh on HepG2 and A549 cells, and the IC_50_ values were 600 μg/mL and 700 μg/mL, respectively. *Sargassum* sp. fractions exhibited in vivo anti-tumor effects in Lewis lung carcinoma cells (LCC) and melanoma B16 cells (MC) [14]. However, as a natural polysaccharide compound, the oral bioavailability of fucoidan is low [15]. Lately, there has been considerable attention paid to polysaccharide breakdown into low molecular weight fucoidan in order to widen its possible utilization [16]. Therefore, it is necessary to carry out controllable hydrogen peroxide degradation to obtain low molecular weight compounds to improve their bioavailability, such that the process will not cause the loss of active groups in the sugar chain [17]. The inhibitory effect of fucoidan degraded by Vc-H_2_O_2_ on A549 cells was stronger than that of natural fucoidan, and the content of L-fucose increased with the decrease of its molecular weight [18]. Different fucoidan showed different inhibitory effects on the same cancer cells, and the sensitivity of different cancer cells to the same fucoidan was not completely consistent. Therefore, the determination of the physicochemical properties of fucoidan and the comparative study of its inhibitory effect on different cancer cells will help to reveal its most significant inhibitory effect on certain cancer cells and the relationship between its structure and activity, and can provide an experimental basis for the development of fucoidans as the next generation of anti-tumor drugs.

In this study, DEAE-cellulose-52 (F1, F2), Vc-H_2_O_2_ degradation, and Sepharose CL-6B gels (DF1, DF2) were used to purify crude fucoidans from *Sargassum*. The physicochemical properties of the four fractions were examined using chemical composition, monosaccharide composition, mean molecular weight (Mw), and FTIR values, and then the structure-activity relationship was analyzed. MTT method was used to comprehensively investigate the cytotoxicity of the four fractions on the human hepatocellular carcinoma cells (HepG2), human burkitt Lymphoma cells (MCF-7), human uterine carcinoma cells (Hela) and human lung cancer cells (A549). Overall, this study will help to screen out cancer cells that are relatively more sensitive to Sargassum fucoidans, after their apoptosis was examined using flow cytometry. To provide experimental basis for the efficient application of Sargassum fucoidans in antitumor drugs.

## 2. Results

### 2.1. Isolation and Purification of Fucoidans

The yield of crude sargassum fucoidan (Fb) were purified by a DEAE-cellulose-52 column (Figure 1). The first peak eluted with distilled water was neutral sugar, and the other two peaks were 0.6 M and 1.5 M NaCl elution peaks, respectively, which were combined and collected and named as F1 and F2. Then, F1 and F2 were dialyzed, concentrated and lyophilized with the yields of 38.57% and 23.61%.

### 2.2. Molecular Weight, Chemical and Monosaccharide Composition of Fucoidans

The contents of polysaccharide, L-fucose, sulfate, uronic acid and the molecular weight were measured and listed in Table 1. The content of polysaccharide (42.82% ± 1.39), L-fucose (13.59 ± 0.54%) and sulfate (18.06 ± 0.32%) of F2 were higher than those of F1. Meanwhile, F2 (562,448 Da) has a larger average molecular weight (Mw) than F1 (32,966 Da), while uronic acid content was lower in F2 (8.24 ± 0.48%) than F1 (9.94 ± 0.90%). After degradation and purification by Separose CL-6B gel column, the average molecular weights of DF1 and DF2 (Figure 2) decreased significantly to 14,893 Da and 21,292 Da. After degradation, the content of polysaccharide and L-fucose of DF1 and DF2 increased, and the sulfate content remained stable or much higher, suggesting that no sulfate was lost during radical degradation. Moreover, the content of polysaccharide (73.93%), L-fucose (23.02%) and sulfate (29.88%) of DF2 were higher than those of DF1.

Table 2 shows the monosaccharide composition of fucoidan F1, F2 and its depolymerized fragments DF1 and DF2. Both fucoidan and its depolymerized fragment had similar neutral monosaccharide constituents. Fucose, mannose, and rhamnose were the main components. After degradation, the contents of glucose and galactose in the degraded fucoidan seemed to decrease. Meanwhile, the content of fucose in fucoidan increased with the purification of fucoidan. The radical process destroyed mostly the branch neutral sugars in sulfated polysaccharides, whereas the sulfated fucose-rich fraction remained unaltered, according to Nardella et al. [19]. Natural and degraded fucoidan contain different amounts of polysaccharide, L-fucose and sulfate, and their monosaccharide compositions are different. Due to the differences in physical and chemical properties between F1, F2, DF1 and DF2, the biological functions of these brown algae polysaccharide extracts need to be further studied.

### 2.3. FTIR Spectrum of Fucoidans

The FTIR spectra (Figure 3) of the degraded fucoidan were comparable to that of the original fucoidan, indicating that degradation did not result in any substantial functional group alterations. The primary distinctive peaks of polysaccharides were observed at 3441–3493 cm^−1^ and 2929–2939 cm^−1^, which corresponds to the stretching vibration of O-H and -CH2, respectively [20]. The two absorption peaks at 1612–1641 and 1408–1420 cm^−1^ corresponded to the symmetric stretching vibration of C=O carbon group and the asymmetric stretching vibration of COO- carboxylic acid group [21], signifying the presence of uronic acids [22]. In addition, the absorption at 1243–1265 cm^−1^ can be attributed to the stretching vibration of S=O, which is consistent with the determination of sulfate groups [20]. The absorption peak intensity can reflect the content of the characteristic functional group [23]. In comparison to the fractions F1 and DF1 (1250 and 1254 cm^−1^), the absorption peak of S=O of F2 and DF2 (1248 and 1243 cm^−1^) was significantly enhanced, consistent with the somewhat higher sulfate group contents (Table 1). The absorption peak at 1033–1049 cm^−1^ is attributed to the stretching vibration of C-O-C and the angular vibration of C-O in the sugar ring C-O-C, while the characteristic absorption peak at about 900 cm^−1^ is caused by β-glycosidic bonds [24]. The absorption at 812–846 cm^−1^ indicates the mode of sulfation. The absorption band at 846 cm^−1^ can be attributed to the equatorial C-O-S bending vibration of the sulfate substituent at the axial C-4 position [25]. The absorption at 812–827 cm^−1^ suggests the presence of 2,4-O-disulfated fucose or 6-O-sulfated GalNAc [26] of the fractions F1, F2 and its depolymerized fragment DF1, DF2, indicating that the sulfation mode did not change significantly. The results showed that the backbone structure of fucoidan was not damaged by oxidation degradation. Consequently, the antitumor activity of degraded polysaccharides may be mainly attributed to the content of polysaccharide and sulfate, as well as molecular weight.

### 2.4. Cell Viability

Inhibition of cancer cell proliferation can be used to evaluate the potential anticancer ability of new substances [27]. In this study, in vitro models (HepG2, A549, Hela, and MCF-7 cells) were used to investigate the anti-cancer properties of native and degraded fucoidans.

As shown in Figure 4, all fucoidans (F1, F2, DF1 and DF2) reduced the viability of HepG2 (Figure 4A) and A549 (Figure 4B) cells in the range of 0.5–8 mg/mL. Interestingly, the fucoidan extracts DF1 and DF2 showed no cytotoxicity to Hela cells at the concentration of 0 to 500 µg/mL (Figure 4C), and the lowest concentration of F1, F2 and DF1 significantly promoted MCF-7 cell proliferation (112.22% ± 5.05%, 118.75% ± 5.10% and 117.98% ± 2.87%) (Figure 4D). In general, DF1 and DF2 exhibited greater cytotoxic effects on the selected cells than that of F1, F2 at high concentrations. These findings suggest that the inhibitory effect of degraded fucoidan DF1 and DF2 on tumor cells (HepG2, A549, Hela, and MCF-7 cells) may be stronger than that of F1 and F2, especially.

Moreover, the IC_50_ value of DF2 against HepG2, A549 and Hela cells was 2.2, 4.9 and 5.9 mg/mL, respectively, and that of DF1 against MCF-7 cells was 7.4 mg/mL (Figure 4E). Compared with A549, Hela and MCF-7 cells, the IC_50_ value was the lowest when HepG2 cells were treated with fucoidan extracts. Specifically, DF2 showed cytotoxicity to HepG2 cells with the lowest half maximal inhibitory concentration value (IC_50_ = 2.2 mg/mL), indicating that DF2 exhibited the greatest cytotoxicity to HepG2 cells and had a better effect on anti-liver cancer.

In addition, similar experiments were conducted using the primary normal human hepatocytes (LO2) to determine whether these fucoidan extracts have toxic effects on normal cells. The results showed that at the concentration of 8 mg/mL, DF2 had the greatest cytotoxicity to LO2 cells, followed by DF1, and F2 had the least cytotoxicity (Figure 4F). Meanwhile, DF2 highly exhibited toxicity to HepG2 cells at concentrations of 1–4 mg/mL with a survival rate of 38.2–61.4%, while the survival rate of LO2 cells was about 74.0–85.4%. Therefore, in vitro anti-hepatoma experiments showed that fucoidan had a strong cytotoxic effect on HepG2 cells and a low cytotoxic effect on normal liver cell LO2, indicating a selective cytotoxicity of fucoidans.

In summary, all tested fucoidan extracts inhibited the growth of HepG2 and A549 cells in the range of 0.5–8 mg/mL, and the inhibitory effect on HepG2 cells was more obvious. DF2 has the strongest cytotoxic effect on HepG2 cells, but has less cytotoxic effect on normal liver cells, so it may be a good potential anti-hepatoma drug.

### 2.5. Flow Cytometry Analysis of Apoptosis of HepG2 Cells

HepG2 cells were treated for 48 h with 0, 1.0, 2.0, and 4.0 mg/mL DF1 and DF2 to reveal if growth suppression is connected to apoptotic induction. To validate the apoptosis produced by Sargassum fucoidans, flow cytometry with Annexin-FITC/PI double labeling was used. Flow cytometry (Figure 5A,B) revealed that HepG2 cells were considerably induced apoptosis after 48 h treatment with DF1 and DF2. When the doses of DF1 and DF2 were 4 mg/mL, the apoptosis rate increased respectively from 6.8% to 39.8% for DF1 and 58.7% for DF2. Moreover, with the increase of DF1 and DF2 concentration, the cell apoptosis rate increased significantly. These findings imply that the suppression of HepG2 cell proliferation can be partially explained by apoptosis induction.

## 3. Discussion

Fucoidan is a water-soluble sulfate heteropolysaccharide with a wide range of biological activities. Fucoidan is most commonly isolated from marine brown algae, exhibits effective inhibitory and killing effects in the modification of tumor cells, and is considered to be the next generation of natural anti-tumor drugs [28]. In this study, two fractions named F1 and F2 were obtained by purification of crude fucoidan extracted in the early stage [29] using DEAE-cellulose-52. F1 (Mw 32,966 Da) and F2 (Mw 562,448 Da) were degraded by Vc-H_2_O_2_, and two degradation products named DF1 (14,893 Da) and DF2 (21,292 Da) were obtained after CL-6B gel purification. The physicochemical structure of initial and degraded Sargassum fucoidan and their anti-tumor inhibitory activity against HepG2, A549, Hela and MCF-7 were evaluated.

F1, F2 and DF1, DF2 had a similar FTIR profile, and all contain the characteristic peaks of functional groups of fucoidan. Additionally, the absorption peaks for S=O (1248 and 1243 cm^−1^) of F2 and DF2 were significantly enhanced, which is consistent with previous results (Table 1). In addition, FTIR showed that the degradation treatment had no significant change in the sulfation mode of polysaccharides, which was consistent with Wang et al. [30]. Therefore, the structures of F1, F2 and DF1 and DF2 were similar, and the increased antitumor activity of the degraded polysaccharides may be mainly attributed to the content of polysaccharide, sulfate and L-fucose, as well as molecular weight.

The chemical composition of the four fractions showed that the contents of polysaccharide and L-fucose in DF1 and DF2 increased compared with those before degradation, and sulfate remained stable or much higher than that before degradation, which is consistent with the results of Li [31] and Wu [32] et al. Meanwhile, there was no obvious loss of sulfate and L-fucose during free radical degradation. In this study, the degraded fucoidan DF1 (Mw 14,893 Da) and DF2 (Mw 21,292 Da) exhibited higher antitumor activity than before degradation. The contents of polysaccharide (42.82%), L-fucose (13.59%) and sulfate (18.06%) of F2 were higher than those of F1, but the molecular weight of F1 (Mw 32,966 Da) was much smaller than that of F2 (Mw 562,448 Da), and the antitumor activity of F1 was higher. Thus, the molecular weight was the most important factor affecting the activity under this condition [33]. While the contents of polysaccharide (73.93%), L-fucose (23.02%) and sulfate (29.88%) of DF2 were higher than those of DF1, and the molecular weights of DF2 (Mw 21,292 Da) and DF1 (Mw 14,893 Da) were similar. Based on the previous study, the sulfate and L-fucose contents were also an important determinant of the antitumor activity [34].

Initial and degraded fucoidans were found to have similar neutral monosaccharide compositions, with fucose, mannose and rhamnose as the main components. The contents of glucose and galactose in the degraded fucoidan seemed to decrease, while the content of fucose increased. This result is consistent with what Zhao et al. have previously reported, that oxidative degradation mainly acts on neutral sugars with low uronic acid and sulfate content, while it had little effect on the degradation of fucoidan with more sulfate radicals bound on the main chain [7].

Studies have found that low molecular weight fucoidan mediates broad-spectrum growth inhibition of human cancer cells, including HeLa cervical adenocarcinoma, HT1080 fibrosarcoma, K562 leukemia, U937 lymphoma, A549 lung adenocarcinoma and HL-60 [5]. In this study, the effects of four Sargassum fucoidan components before and after degradation on HepG2 (Figure 4A), A549 (Figure 4B), Hela (Figure 4C) and MCF-7 (Figure 4D) tumor cells were evaluated. Among them, the anti-tumor effects of degraded fucoidan with low molecular weight (DF1 and DF2) were higher than those of natural fucoidan (F1 and F2). All Sargassum fucoidan components increased the in vitro cytotoxicity of HepG2 and A549 cells in a dose-dependent manner between 0.5 and 8.0 mg/ml. The lowest IC_50_ values of HepG2, A549, Hela and MCF-7 cells were 2.2 (DF2), 4.9 (DF2), 5.9 (DF2) and 7.4 (DF1) mg/mL, respectively (Figure 4E). The IC_50_ value of HepG2 cells exposed to fucoidan components reached the minimum, and the IC_50_ value of DF2 to HepG2 was the lowest (IC_50_ = 2.2 mg/mL). The apoptosis of HepG2 cells was detected by flow cytometry, and it was found that both DF1 and DF2 could induce obvious apoptosis of HepG2 cells with increasing concentration. In HepG2 cells, the percentage number of apoptosis produced by DF2 (46.25% ± 3.62) was substantially larger than that of DF1 (33.15% ± 1.84). Sairong Fan et al. [35] has demonstrated that polysaccharide SFPS derived from Sargassum fusi-forme (Harv.) Setche may induce apoptosis in HepG2 cells. Therefore, the current study shows that the fucoidan of Sargassum has a significant effect on inhibiting liver cancer cells, and the fraction DF2 has the best anti-liver cancer effect. Cong et al. [36] isolated and prepared sulfated polysaccharides 04S2p-s and Alg-S from Sargassum, and tested their anti-tumor activity on five different tumor cell lines. Alg-S showed significant anti-tumor effect on Bel7402, SMMC7721 and HT-29 cell lines, while 04S2P-S only showed significant anti-tumor effect on Bel7402 cell line. Shao et al. [12] examined the cytotoxicity of SHP30, SHP60 and SHP80 on human colorectal cancer DLD cells. SHP30 and SHP60 showed significant inhibitory effects, especially at the concentration of 4.0 mg/mL, where the inhibition rate was the highest, 85.3% and 76.37%, respectively. In addition, 6.0 mg/mL of SHP80 had the highest inhibition rate of 44% on DLD cancer cells. It was proved that the effects on proliferation activity of the same cell differed by substance type and its dose, and the sensitivity of different cells to the same compound was not completely consistent. Nevertheless, knowledge about the differential inhibitory effects of fucoidan on different cancer cells is limited. Fucoidan is a multifunctional molecule that interacts with various cancer-related cellular signaling pathways. Some signaling pathways can affect one or more processes of apoptosis, cell cycle arrest, anti-angiogenesis, anti-metastasis, anti-oxidation and immune regulation [37]. Suresh et al. [13] found that the same fucoidan component SP extracted from Sargassum showed different anti-proliferative mechanisms against HepG2 and A549 cancer cells. Therefore, further experiments are needed to investigate specific pathways on fucoidan-induced cancer cell death, especially in vivo studies, to elucidate the underlying mechanisms behind these differences in tumor inhibition.

In general, all tested fucoidan fractions showed inhibition on the growth of HepG2 and A549 cells, with strong cytotoxicity on HepG2 cells and low cytotoxicity on normal liver cell LO2. Flow cytometry also revealed that Sargassum fucoidan might trigger apoptosis in human hepatocellular carcinoma cells (HepG2). Therefore, fucoidan from Sargassum may be a potential anti-hepatoma drug. The biological function of fucoidan is controlled by its molecular structure, such as its focal bond, sugar type, sulfate content and molecular weight [33], and molecular weight seems to be the most important determinant [38]. DF2, the degradation fucoidan fraction of Sargassum with a low molecular weight and the highest content of polysaccharide, sulfate and L-fucose, had the most obvious inhibitory effect on HepG2 tumor cells. Therefore, we can also speculate that the increase of antitumor activity of fucoidan may be related to the molecular weight and the content of polysaccharide, sulfate and L-fucose. The results of this study will provide a theoretical reference to degraded fucoidan as a natural chemical preventive agent for the adjuvant treatment of cancer, especially for liver cancer.

## 4. Materials and Methods

### 4.1. Materials

In our previous study, crude fucoidans were extracted from *Sargassum hemiphyllum* (Turn.) C.Ag. by ultrasonic-assisted hot water extraction, ethanol fractional precipitation and Sevage method [29].

The human hepatocellular carcinoma cells HepG2, human burkitt Lymphoma cells MCF-7, human uterine carcinoma cell line Hela and human lung cancer cell line A549 were obtained courtesy of the Cell Bank of the Chinese Academy of Sciences, Shanghai. The primary human normal hepatocytes LO2 were purchased from the ATCC Cell Line Bank, USA. Dulbecco’s Modified Eagle Medium (DMEM, Catalog no. 11995-500), fetal bovine serum (FBS, Catalog no. 10270-106), penicillin-streptomycin (Pen-Strep, Catalog no. 15140-122), trypsin-EDTA (Catalog no. 25200-072) and phosphate buffered solution (PBS, Catalog no. 10010-023) were purchased from Gibco of the Thermo Fisher Scientific, Shanghai, China. Methyl thiazolyl tetrazolium (MTT, Catalog no. 298-93-1), Dextran (Catalog no. S14102, S14107, S14109, S14113, S14115), the standards (Mannose, rhamnose, glucose, galactose, xylose, arabinose, L-fucose), DEAE-cellulose DE-52(DEAE-52, Catalog no. S14024), Sepharose CL-6B (CL-6B, Catalog no. S14087) were purchased from Yuanye Biology Science and Technology Company, Shanghai, China. Dimethyl sulfoxide (DMSO, Catalog no. 2206-27-1) and aqueous phenol solution (Catalog no. R012316) were obtained from Eon Chemical Technology company, Shanghai. A cell Cycle kit (Catalog no. C1052) was purchased from Biyuntian Biotechnology company, Shanghai, China. An apoptosis kit (item No. 211-01/02) was purchased from Novizan Biotechnology company, Nanjing, China.

### 4.2. Purification of Fucoidan from Sargassum hemiphyllum (Turn.) C.Ag.

The crude polysaccharide Fb was purified using the DEAE-cellulose-52 column described by Guo et al. [39]. Briefly, Fb (0.1 g) was dissolved in 100 mL of distilled water and loaded on a DEAE-cellulose-52 column (3.5 × 40 cm, Cl^−^ form). Initially, elution was carried out with distilled water, followed by stepwise elution with increased concentrations of NaCl (0.2–1.8 M) at a flow rate of 1 mL/min. The eluent was collected in test tubes using an automated step-by-step fraction collector. The content of polysaccharides of each tube was measured at 490 nm by the phenol-sulfuric acid colorimetric method until complete elution. The elution fractions of 0.6 M and 1.5 M were collected and named as F1 and F2, respectively, and then dialysis, concentration and lyophilization.

### 4.3. Degradation of Fucoidan

Degraded low-molecular-weight fucoidan (F1 and F2) were obtained using hydrogen peroxide and ascorbic acid according to partially modified method by Deng et al. [40]. In brief, the powder (0.3 g) was dispersed by distilled water (1.5%, *w*/*v*) in a thermostatic water bath tank (30 °C). Then, ascorbic acid (0.2072 g) was added to the solution, and hydrogen peroxide (5%, 1 mL) was added while stirring. The mixture was produced by centrifugation following two hours of magnetic stirring. The supernatant was dialyzed, concentrated, and freeze-dried. Finally, the powder was purified using a Separose CL-6B gel column (1.6 × 60 cm) and lyophilized to yield DF1 and DF2.

### 4.4. Chemical Composition Analysis

The polysaccharide content was determined using the phenol-sulfuric acid colorimetric technique, and L-fucose was used as the standard [41]. The L-fucose content was analyzed by the method of Dische colorimetric using L-fucose as the standard [42]. A modified carbazole colorimetry was used to calculate uronic acid [43]. The sulfate concentration was measured by BaCl_2_ gel method [44].

### 4.5. The Average Molecular Weight (Mw) Determination

Molecular weights were determined on a high-performance liquid chromatograph (Agilent 1100, Santa Clara, CA, USA) with a gel-filtration chromatographic column of Ultrahydrogel- TM500 (7.8 × 300 mm, Waters, Milford, MA, USA). The mobile phase was a 0.2 M Na_2_SO_4_ aqueous solution flowing at 0.6 mL/min. All the samples were filtered through 0.45 μm filter membrane before being analyzed. The detection was carried out at 35 ℃ with a refractive index detector (Agilent 1200, Santa Clara, CA, USA). Six different molecular weight dextrans (200, 100, 20, 40, 10 and 4 kDa) were used as the standard.

### 4.6. Monosaccharide Composition Analysis

The monosaccharide compositions were determined using PMP derivatization and HPLC [45]. Briefly, 5 mg of polysaccharides was hydrolyzed with 1 mL of 2 M trifluoroacetic acid (TFA) under 110 °C for 6 h. After cooling to room temperature, TFA was eliminated using a rotary evaporator and repeated additions of methanol (2 mL). Thereafter, 1 mL of ultra-pure water was added to create the polysaccharide solution. Then, a 5 mL glass tube with plug was filled with 400 µL of polysaccharide solution, 5 mg/mL of monosaccharide standard solution (Man, Rha, Glu, Gal, Xyl, ara, and Fuc), and mixed monosaccharide standard solution, respectively. After that, 400 μL sodium hydroxide solution and PMP derivative reagent were added in turn and reacted at 70 °C for 100 min. After cooling to room temperature, 400 μL hydrochloric acid was added to neutralize, and then extracted with 2 mL chloroform. The supernatant was filtered through a 0.22 μm aqueous membrane and loaded into a sample bottle for use. Analysis was conducted on an HPLC system (Waters E2695, Milford, MA, USA) with an analytical Kromasil 100-5-C18 column (150 × 4.6 mm, 5 µm, Kromasil M05CLA15, Göteborg, Sweden). The mobile phases were 0.05 M phosphate buffer solution (pH 6.8) and acetonitrile (83: 17, *v*/*v*), with a flow rate of 1 mL/min, 30 ℃. The UV detection wavelength was set at 245 nm.

### 4.7. Fourier Transformed-Infrared (FTIR) Spectrometric Analysis

According to Jia’s procedure [46], the FTIR analysis was completed. The dried fucoidan sample was mixed with potassium bromide (KBr). The mixture was grinded in an agate mortar for 5–10 min, and then pressed into tablets, which were scanned by FTIR (Bruker Tensor-27, Billerica, Massachusetts, Germany) in the range of 500–4000 cm^−1^ with KBr as background.

### 4.8. In Vitro Anti-Tumour Activity Assay

#### 4.8.1. Cell Culture

According to the approach of Cao et al. [47], HepG2, MCF-7, Hela, A549 and LO2 cells were cultured at 37 °C in a 5% CO_2_ incubator, and maintained in Dulbecco’s Modified Eagle’s medium (DMEM) supplemented with 10% (*v*/*v*) fetal bovine serum and 1% (*v*/*v*) penicillin and streptomycin. The medium was renewed every 2 days to maintain them in an exponential phase.

#### 4.8.2. Cell Viability

An MTT assay was used to evaluate the cellular viability as previously described [48]. Briefly, HepG2, MCF-7, Hela and A549 cells were seeded in 96 well tissue culture plates at a density of 2 × 10^3^ cells/well. LO2 cells were seeded at a density of 5 × 10^3^ cells/well. After 24 h, cells were treated with 100 μL fucoidan at concentrations of 2, 4 or 8 mg/mL for an additional 48 h. At the end, 20 μL MTT (5 mg/mL) was added to each well and incubated for another 4 h at 37 °C in a 5% CO_2_ incubator. After removal of the supernatants, 150 μL of DMSO was added to each well and mixed for 10 min. Next, the absorbance was measured at 570 nm using a VarioskanFlash microplate reader (Thermo Fisher Scientific, Waltham, MA, USA).

The following equation was used to calculate the inhibition rate:Cell Inhibition rate (%) = (D_i_ − D_0_)/(D_j_ − D_0_) × 100%(1)

In Equation (1), D_0_ was the absorbance of the blank, D_i_ was the absorbance in the presence of samples, and D_j_ was the absorbance of the control.

### 4.9. Cell Apoptosis Experiment

HepG2 cells were seeded in 6-well culture plates (1.2105 cells/well) and treated for 48 h with 0, 1.0, 2.0, and 4.0 mg/mL DF1 and DF2. After that, the cells were collected by centrifugation at 1000 rpm. Following three washes with PBS, the cells were resuspended in 100 L Binding Buffer. The cells were then stained at room temperature for 10 min with 5 µL Annexin V-FITC (FITC) and 5 µL Propyl iodide (PI), and mixed gently with 400 μL Binding Buffer. At the end of the staining process, apoptosis rates were measured by Cyto FLEX flow cytometry (Beckman Coulter, Krefeld, Germany).

### 4.10. Statistical Analysis

All the experiments were repeated at least three times. Values were expressed as means ± standard deviation (SD). Significant difference between groups were determined using One-Way ANOVA followed by LSD multiple comparison, and the differences were considered to be statistically significant at *p* < 0.05. All analyses were conducted using the statistical software, SPSS 26.0.

## 5. Conclusions

In this study, four fucoidans (F1, F2, DF1 and DF2) were purified and degraded from crude fucoidans of *Sargassum*, physicochemically characterized, and tested for antitumor activity in vitro. It was discovered that fucoidan’s molecular weight reduced after degradation even though its structure and chemical composition were different before and after degradation. Under the same conditions, the inhibitory rate of the fucoidan DF2, which had a lower molecular weight and the greatest concentration of polysaccharide, sulfate, and fucose, increased to 71.63% after 48 h. The IC_50_ of DF2 against HepG2, MCF-7, Hela, and A549 tumor cells was 2.2 mg/mL, which was lower than the IC_50_ against other components. Moreover, Flow cytometry demonstrated that HepG2 hepatocellular carcinoma cells might undergo apoptosis when exposed to fucoidan of Sargassum. Therefore, the antitumor activity of Sargassum fucoidan was related to its L-fucose, sulfate content and molecular weight, with the most significant inhibitory effect being on HepG2 hepatocellular carcinoma cells. These results suggest that Sargassum fucoidan has anti-hepatocarcinogenic potential, but its potential conformational relationship and anti-tumor mechanism need to be further investigated.

## Figures and Tables

**Figure 1 molecules-28-02610-f001:**
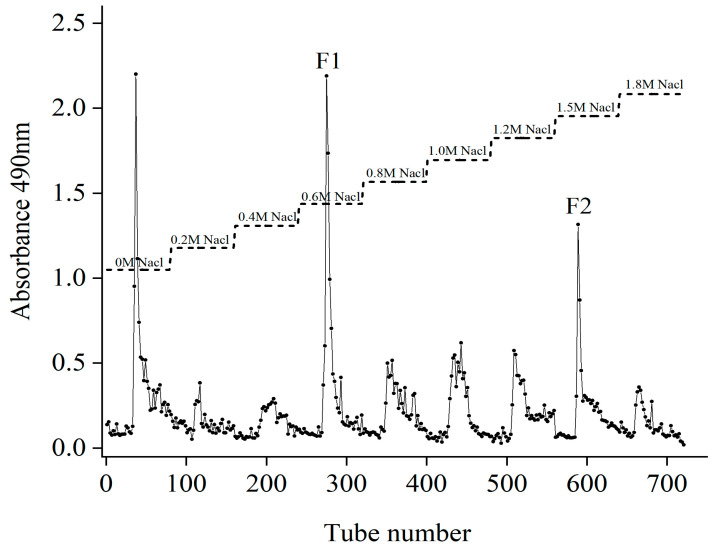
Purification of polysaccharides by DEAE- anion exchange chromatography.

**Figure 2 molecules-28-02610-f002:**
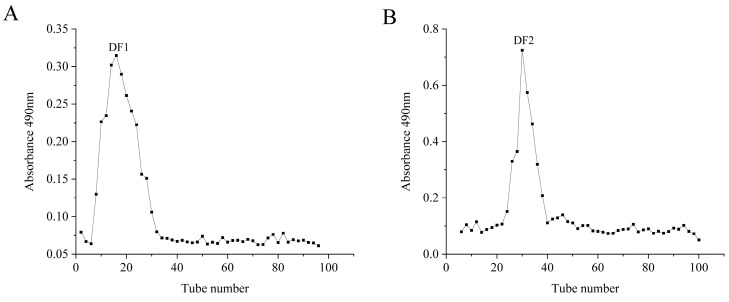
CL-6B gel column elution curve of DF1 (**A**) and DF2 (**B**).

**Figure 3 molecules-28-02610-f003:**
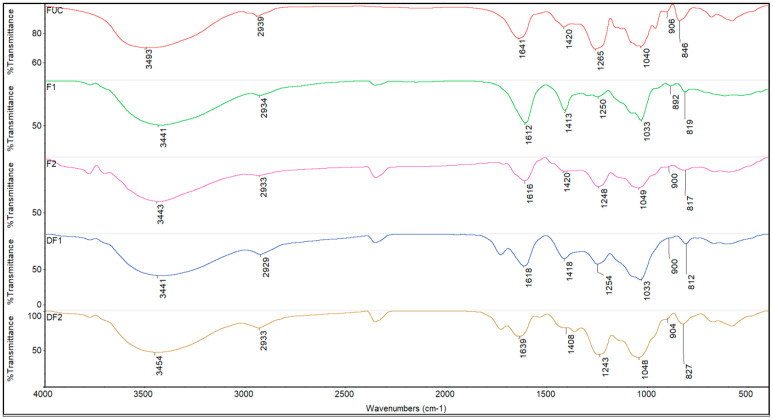
The IR spectra of standard fucoidan (FUC), fractions F1, F2 and its depolymerized fragment DF1, DF2.

**Figure 4 molecules-28-02610-f004:**
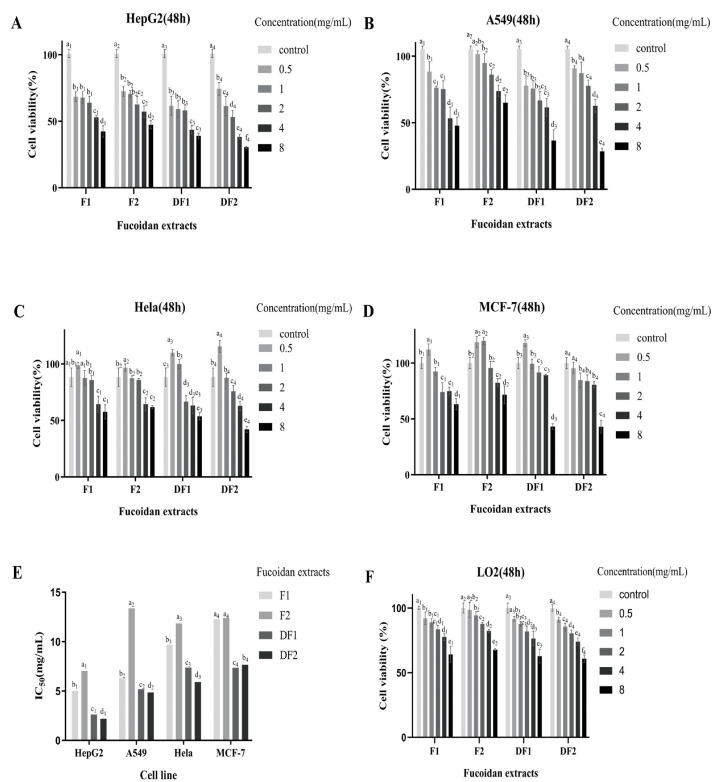
Effects of F1, F2, DF1 and DF2 on cell viabilities of HepG2 (**A**), A549 (**B**), Hela (**C**), MCF-7 (**D**) and LO2 cells (**F**). Cell viability was evaluated after 48 h of coincubation with varied doses of F1, F2, DF1, and DF2. Bar plot (**E**) show the half maximal inhibitory concentration (IC_50_) values (the inhibitory concentrations at 50% growth of HepG2 (**A**), A549 (**B**), Hela (**C**) and MCF-7 (**D**) cells) of F1, F2, DF1 and DF2. Different letter in bar plot indicates significant difference among the same fucoidan treatment group (*p* < 0.05). Each experiment was repeated three times.

**Figure 5 molecules-28-02610-f005:**
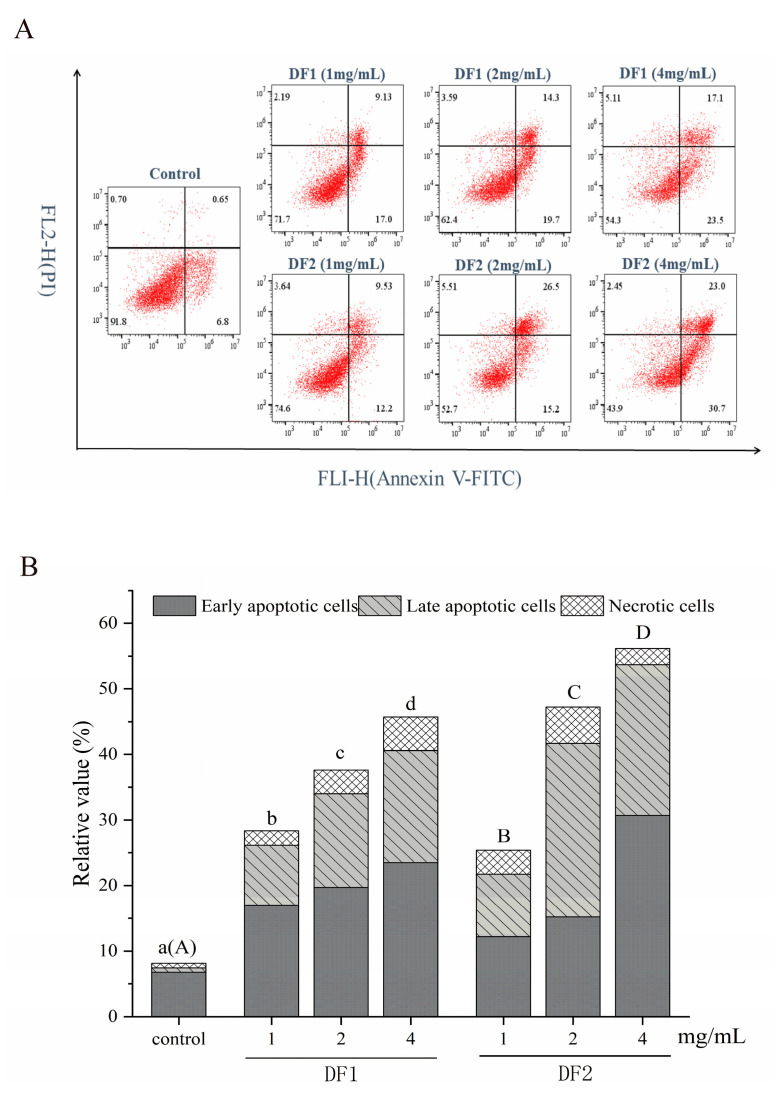
The effect of DF1 and DF2 on HepG2 cell apoptosis. (**A**) Representative scatter plot of Annexin V/PI staining. HepG2 cells were treated with DF1 and DF2 (0, 1.0, 2.0, and 4.0 mg/mL) for 48 h before being stained with Annexin V-FITC/PI solution and identified by flow cytometry. The upper right quadrants and lower right quadrants represent the ratios of late apoptotic cells early apoptotic cells, respectively. (**B**) Apoptotic cell histogram. The distinct letters of the same component exhibited significant differences at the 0.05 level. All tests were done three times.

**Table 1 molecules-28-02610-t001:** Chemical components and Mw of fucoidans.

Samples	Polysaccharide (%)	L-Fucose (%)	Sulfate (%)	Uronic Acid (%)	Mw (%)
F1	30.62 ± 2.44	11.00 ± 0.57	13.60 ± 1.45	9.94 ± 0.90	32,966
F2	42.82 ± 1.39	13.59 ± 0.54	18.06 ± 0.32	8.24 ± 0.48	562,448
DF1	60.90 ± 4.35	12.67 ± 0.98	13.53 ± 0.60	32.99 ± 1.41	14,893
DF2	73.93 ± 2.35	23.02 ± 1.13	29.88 ± 1.09	15.07 ± 0.97	21,292

**Table 2 molecules-28-02610-t002:** Monosaccharide composition of fucoidans.

Samples	Sugar (%)					
	Man	Rha	Glu	Gal	Xyl	Fuc
F1	22.82	18.61	18.72	15.77	ND	24.00
F2	22.81	24.34	10.85	8.17	7.81	26.03
DF1	20.79	20.36	9.17	9.10	7.54	33.02
DF2	19.33	32.94	2.95	ND	ND	44.78

Note: Based on liquid chromatograms, the percentage was computed by considering the overall area of the peaks as 100%. Man, mannose; Rha, rhamnose; Glu, glucose; Gal, galactose; Xyl, xylose; Fuc: fucose, ND, not detectable.

## Data Availability

Not applicable.

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
