# Peer review of "Physicochemical Characterization and Antitumor Activity of Fucoidan and Its Degraded Products from Sargassum hemiphyllum (Turner) C. Agardh"

_molecules, 2023, doi:10.3390/molecules28062610_

Round 1

Reviewer 1 Report

The concept of the study is basic and the draft has written according to the journal's format, however some more fuzzy statements are quoted in this article and seems appropriate references are not given. The specific comments are appended here for correction before publication. I recommend the manuscript can be accepted after minor revision.

1) The author must include current (2023) references in this manuscript.

2) Did the author purify the fucoidan through DEAE cellulose, Why? Is this purify or partially purified

3) Discussion part must be rewrite, dually compared with standard references.

4) What is the main motto of degradation work here?

5) Conclusion part is not clear, must be improve.

Reviewer 2 Report

 In this study, the authors  characterized  and tested the cytotoxicity of fucoidan from Sargassum hemiphyllum (Turner) C. Agardh. They tested the anti-proliferative effects of purified fucoidans on HepG2, MCF-7, Hela and A549 cell lines by MTT.

Overall, it is well-established study at its chemical part. However, very poor at the biological part.

Major comments

  1. The cytotoxicity IC50 of the Fucoidan fractions is very high (1 to 12 mg/ml). Others have reported much lower IC50 values (tens of micrograms/ml). Does such high concentration can be considered as pharmacological? To which extend did the viscosity of the cell media changed when treated with such high concentration of poly sugar? How about the osmotic shock?
  2. The authors have tested the effect of the Fucoidan fractions on cancerous cell lines only. They must test the same fractions cytotoxicity on two normal cell lines (preferably related to the same organs sources of the cancerous ones).
  3. MTT is NOT a good model for any specific toxic pathway in the cell. The authors must test the effect of the Fucoidan fractions on specific programmed cell deth like apoptosis.

Reviewer 3 Report

The idea of this article is novel. But there are weaknesses in some parts of this manuscript. They are listed below:

In the results section, first, state the results and then display the figure or table.

The IC50 value is related to the substance, in one of the sentences you attributed it to the cell. Please correct the sentence.

Could you discuss further the possible reasons for the difference in the bioactivity of polysaccharides?

The purpose of the degradation of fucoidan is not well-defined in the manuscript. Could you explain a little more?

The conclusion can be written in more detail.

Round 2

Reviewer 2 Report

Accepted